# Effect of Sucrose on Growth and Stress Status of *Castanea sativa* x *C. crenata* Shoots Cultured in Liquid Medium

**DOI:** 10.3390/plants11070965

**Published:** 2022-04-01

**Authors:** Diego Gago, María Ángeles Bernal, Conchi Sánchez, Anxela Aldrey, Beatriz Cuenca, Colin Bruce Christie, Nieves Vidal

**Affiliations:** 1Misión Biológica de Galicia Sede Santiago de Compostela, Consejo Superior de Investigaciones Científicas, Apdo 122, 15780 Santiago de Compostela, Spain; tresdedecembro@gmail.com (D.G.); conchi@iiag.csic.es (C.S.); anxela.aldrey@iiag.csic.es (A.A.); 2Departamento de Biología, Facultad de Ciencias, Universidade da Coruña, Campus da Zapateira s/n, 15071 A Coruña, Spain; angeles.bernal@udc.es; 3Maceda Nursery, Tragsa-SEPI Group, Carretera de Maceda a Baldrei km 2, 32700 Maceda, Spain; bcuenca@tragsa.es; 4The Greenplant Company, Palmerston North 4410, New Zealand; brucechristie101@gmail.com

**Keywords:** bioreactors, chestnut, continuous immersion, photoautotrophy, photosynthesis, temporary immersion

## Abstract

Current breeding programs aim to increase the number of ink-tolerant chestnut trees using vegetative propagation of selected genotypes. However, the commercial vegetative propagation of chestnut species is still a bottleneck for the forest industry, mainly due to problems in the rooting and acclimation of propagules. This study aimed to explore the potential benefits of decreasing sucrose supplementation during chestnut micropropagation. Explants were cultured with high light intensity and CO_2_-enriched air in temporary or continuous immersion bioreactors and with different sucrose supplementation to evaluate the impact of these treatments on growth, rooting and physiological status (monosaccharide content, soluble phenolics and antioxidant activity). The proliferation and rooting performance of shoots cultured by continuous immersion decreased sharply with sucrose concentrations lower than 1%, whereas shoots cultured by temporary immersion grew and rooted successfully with 0.5% sucrose. These results suggest this system is appropriate to culture chestnut with low sucrose concentration and to explore photoautotrophic propagation of this species.

## 1. Introduction

European chestnut (*Castanea sativa* Mill., also known as sweet chestnut) is a long-lived multipurpose tree cultured worldwide for its nuts and timber [1]. Its by-products contain potentially valuable bioactive compounds with antioxidant, anticarcinogenic and cardioprotective properties [2]. In countries with a long tradition of cultivation, agroforestry systems of *C. sativa* have special forest structures, which provide specific ecosystem services such as the provision of food (for people and wildlife), materials and energy, climate regulation by carbon storage and wind, temperature and humidity moderation at a local level, and in a more generic sense, erosion regulation and the conservation of traditional knowledge, as well as landscape value and recreational uses [3]. The ecosystem services concept has become increasingly important in environmental science and management during recent decades because of the connection and relationships between biodiversity, ecological functioning and human well-being [4]. The ecosystem services provided by chestnut agroforest systems maintain biodiversity and can help mitigate climate change [3].

Currently, chestnut trees are threatened in Europe because of ink disease caused by several species of the hemibiotrophic oomycete *Phytophthora*, such as *P. cinnamomi* and *P. cambivora* [5]. These pathogens spread mainly through the movement of soil harboring inoculum and the dissemination of asexual flagellated spores that can actively travel short distances or passively travel long distances in flowing water [6,7]. During the initial biotrophic period, it may obtain nutrients from living plant tissues causing minimum damage to the plant, but during the hemibiotrophic phase it invades the host’s roots [8], which leads to subcortical necrosis of the root system and the basal part of the stem; this is followed by the appearance of wasting symptoms in the foliage until the total desiccation and death of the tree [9,10,11]. Given its dependence on soil water, this pathogen is potentially very damaging in a climate change scenario, as abiotic stresses such as drought, flooding, rising temperature and moisture stress predispose trees to infection [12]. Breeding programs have produced disease-tolerant trees by crossing susceptible *C. sativa* with resistant Asian species *C. crenata* and *C. mollissima* [13]. However, hybrids obtained from controlled crosses and trees originated through spontaneous hybridization are not well adapted to the main areas of chestnut cultivation because they lack the adaptation to environmental stresses, summer droughts, flooding and winter chilling [14]. In addition, many of them do not show the typical architecture of sweet chestnut, the fruit characteristics, such as flavor, and the ripening period or graft compatibility with traditional *C. sativa* varieties.

This highlights an important environmental, social and economic problem that requires new selection strategies to produce disease-tolerant chestnut trees with a larger presence of *C. sativa* genes that are able to thrive when challenged by *Phytophthora* in the affected areas. Several programs aimed at selecting new chestnut material, better adapted to the current and predicted future climatic conditions, have been developed in Europe [15,16,17,18].

These selected genotypes need to be propagated vegetatively to maintain their tolerance to the disease. As chestnut is highly recalcitrant to conventional vegetative propagation, agar-based micropropagation protocols have been developed [19,20]. Despite recent progress [21,22,23], the commercial vegetative propagation of chestnut species is still a bottleneck for the forest industry, mainly due to problems in rooting and acclimation (reviewed in Vielba et al. [1]). Actions aiming to increase root quality and acclimation efficiency should be directed to improve both shoot and root performance.

Micropropagated plants are frequently cultured in closed containers in photomixotrophic conditions: low light intensity, limited CO_2_ availability and using sugars as an energy and carbon source. The explants are submitted successively or simultaneously to environmental conditions, which differ from those found in nature and can cause oxidative stress, such as wounding, abnormal mineral nutrition, hormonal treatments, high relative humidity and possible accumulation of different gases such as ethylene in the confined atmosphere, a rupture of the usual cohabitation with the epiphytic microorganisms and osmotic perturbances due to the sucrose content of the medium [24].

Bioreactors with forced ventilation can eliminate some of these problems and thereby enhance the physiological characteristics of micropropagated plants [25]. This technology has been successfully applied to a range of woody species [26]. In our laboratory, we developed the first protocols for culturing chestnut shoots in liquid medium using forced ventilation, either by temporary immersion (TIS) [27] or by continuous immersion systems (CIS) [28]. In both cases, 3% sucrose was added to the medium as a source of carbon and energy.

High sucrose concentrations may cause hyperhydricity and other disorders [29]. It has been claimed that the elimination of sugar from micropropagation nutritive media promotes photosynthetic activity, producing a healthier physiological state and increasing the speed of plantlet adaptation to greenhouse conditions [30]. However, lowering sucrose supplementation alone usually reduces the growth of the explants, and for this reason most of the laboratories still add sucrose to the culture medium, as a source of carbon and energy. To counterbalance the effects of decreased sucrose supply on plant metabolism, it is necessary to create conditions that enhance the photosynthetic activity inside the culture vessel, in particular: high light intensity and high CO_2_ [30,31,32].

This study aimed to explore the possible benefits of decreasing sucrose supplementation during chestnut micropropagation. Previously, we defined the basic requirements to culture chestnut by CIS in photoautotrophic conditions, namely, high light intensity, forced ventilation with CO_2_-enriched air and explants with well-developed leaves [33,34]. However, the proliferation rates in a sucrose-free medium were low. In addition, survival was only possible after two or three serial subcultures with decreased sucrose concentrations, indicating the need to adapt the physiological status of the shoots to the new conditions. In this study, we compared explants cultured in different bioreactors and with different sucrose supplementation to evaluate their impact on growth, rooting and physiological status.

## 2. Results

### 2.1. Proliferation

Typical shoot growth from clones CO53 and PO42 cultured in liquid medium supplemented with 0.5, 1 and 3% sucrose is shown in Figure 1 and Figure 2 (below). All treatments produced shoots suitable for subculturing, but sucrose supplementation affected differently the shoots cultured by TIS and by CIS.

In clone CO53, cultured by CIS, sucrose supplementation had a significant effect in all the studied growth parameters (*p* = 0.009 for NS and *p* < 0.001 for SL, LW and LL). The lowest concentration of sucrose (0.5%) produced significantly fewer shoots than the other two treatments, and these shoots were smaller and had smaller leaves (Figure 3). The best treatment was 1% sucrose, which yielded significantly longer shoots and leaves than the conventional sucrose concentration (3%).

In clone PO42, cultured by TIS, all treatments produced similar proliferation, shoot and leaf qualities, although a small increase in shoot size was observed when less sucrose was added to the medium (Figure 4). Non-significant differences were found for NS, SL and LL (*p* = 0.790, 0.245 and 0.467, respectively). Reduced sucrose concentration only affected leaf width (*p* < 0.001) significantly, with narrower leaves in the 0.5% and 1% sucrose treatments.

### 2.2. Rooting

In the experiments shown in Table 1 and Figure 5, we studied how the rooting response was affected by the sucrose supplementation during (a) the previous multiplication phase and (b) the rooting expression phase. For the two clones, the highest sucrose content enhanced rooting and the best results were obtained when both the multiplication and the rooting phase were carried out with 3% sucrose (Table 1, Figure 5a,c). However, differences between the two genotypes were detected. In clone PO42, rooting percentages ranged from 56 to 97% and shoots cultured in 0.5% sucrose obtained a 72% rooting percentage when 1% sucrose was added to the rooting media (Figure 5d). In this clone, the differences in rooting were mainly due to the sucrose content of the proliferation medium, whereas the sucrose content during rooting had a smaller contribution.

In the clone CO53, cultured by CIS, less shoots formed roots, with percentages ranging from 6 to 84%. The lowest response (6%) corresponded to shoots proliferated with 0.5% sucrose and rooted with the same sucrose concentration (Table 1, Figure 5b). This value contrasts with the response of clone PO42, cultured by TIS, to the same treatment (66%). In CO53, the sucrose content of both the multiplication medium and the rooting medium influenced rooting success, indicating that changes in either stage allow modulating the rooting response to some extent: the use of 0.5% sucrose for rooting was clearly detrimental (with percentages ranging from 6 to 25%, but shoots cultivated in 0.5% sucrose showed an acceptable rooting response (56%) when the rooting medium was supplemented with 1% sucrose.

### 2.3. Biochemical Analyses

The content of total monosaccharides of the leaves of PO42 and CO53 shoots harvested after 8 weeks in the proliferation medium is shown in Figure 6. Monosaccharide content was positively correlated with the amount of sucrose supplemented to the culture media (*p* < 0.001). Leaves from clone PO42 cultured by TIS showed higher sugar content than leaves from CO53 cultured by CIS (*p* < 0.001), and in the former, the differences between the 0.5 and 3% treatments were almost three-fold smaller than those observed in CO53.

As shown in Figure 7, the antioxidant activity of CO53 and PO42 shoots significantly decreased when more sucrose was added to the medium (*p* < 0.001); the values observed in shoots cultured with 3% sucrose were about three times smaller than with 0.5% sucrose. Both clones followed the same trend, but in clone CO53, the antioxidant activity was higher than in clone PO42 (*p* = 0.009).

Soluble phenolic compounds (Figure 8) were apparently less affected by sucrose supplementation than the morphological and biochemical parameters previously described. Similar values were observed between treatments (*p* = 0.147), although clone CO53 shoots had significantly more soluble phenols than PO42 (*p* = 0.029).

## 3. Discussion

Our study demonstrates the feasibility of propagating chestnut by TIS and CIS with the sugar supplementation reduced to one-sixth of the normal 3% used for this species. These results differ from those reported by Sáez et al. [35], who concluded that to increase PPF from 50 to 150 µmol photons m^−2^ s^−1^ did not increase photosynthesis enough to sustain chestnut growth when sucrose was reduced from 3 to 0.5%. These authors cultured chestnut shoots in tubes with a semisolid medium and without forced ventilation, and in these conditions shoot growth is frequently sucrose-dependent, as shown for kiwifruit (*Actinidia deliciosa*) [36], *Arabidopsis* [37], *Pfaffia glomerata* [38] and *Vernonia condensata* [39]. In a recent study we successfully micropropagated willow under high PPF and without sucrose when we used bioreactors ventilated with CO_2_-enriched air instead of jars without ventilation [40], and to obtain good proliferation in chestnut we applied the same principle, adding CO_2_-enriched air every 90 min to the bioreactors. Increased CO_2_ concentration inside the flasks enabled shoot growth of other plants in media with low sucrose or without any supplementary carbohydrate, as reported for *Paulownia fortunei* (Seem.) Hemsl. [41], *Samanea saman* (Jacq.) Merr. [42], several *Eucalyptus* species [43,44,45,46] and *Macadamia tetraphylla* [47].

In this study with chestnut, we used two propagation systems with liquid media: temporary and continuous immersion. The two genotypes in the study (CO53 and PO42) had been successfully cultured in both systems with 3% sucrose [27,28] and by CIS during the first attempts to micropropagate chestnut in photoautotrophic conditions [33,34]. Within each system, the growth responses of both clones were similar, and for this reason we hypothesize that the differences in the response to sugar supplementation detected in this study may be more related to the culture system (TIS or CIS) than to the genotype.

Sucrose reduction affected proliferation and rooting differently in TIS and CIS. In TIS (clone PO42), the reduction in sucrose concentration to 0.5% did not affect the growth parameters during proliferation. By contrast, in CIS (clone CO53), the explants cultured with 0.5% sucrose produced fewer and smaller shoots than those cultured with 1 and 3% sucrose, although in all treatments shoot number and growth enabled multiplication of the plant material. The best treatment for proliferation in CIS was 1% sucrose, with longer shoots than those produced by 3% sucrose in CIS and also longer than shoots produced by TIS.

Regarding rooting, sucrose treatments had a clear influence in both culture systems: in TIS, shoots rooted better when they had been proliferated with higher sucrose concentrations, and in CIS, the sugar content during both proliferation and rooting had an effect on root formation. It has been reported that plants grown with low levels of sucrose develop higher photosynthetic activity and root better [30]. In potato, tobacco [29], thyme [32], *paulownia* [41], papaya [48] and wasabi [49], more roots were formed when less sugar was in the culture medium. However, in our study with chestnut, we observed a positive correlation between sucrose supplementation and rooting. Similar responses have also been reported in other plants such as grapevine [50], kiwifruit [51], Chinese ash [52], poplar [53], and willow [40].

Adventitious rooting is an energy-dependent process that needs to supply carbohydrates to the root generation region [54,55]. Auxin stimulates the mobilization of carbohydrates in leaves and the upper stem and increases translocation towards the rooting zone [56]. Low carbohydrate levels in cuttings at the beginning of rooting limit the speed or number of roots formed [57], although the application of sugars to the rooting medium can increase the subsequent root formation [58,59]. In our study, shoots collected at the end of the proliferation phase showed different monosaccharide content depending on whether they had been cultured by CIS or by TIS. Shoots cultured by CIS showed lower rooting percentages, lower monosaccharide content and more pronounced differences in these sugars between sucrose treatments, possibly indicating sugar availability could be a limiting factor for root formation in these shoots. The shoots cultured with 0.5% sucrose had the lowest monosaccharide content and the lowest rooting response. This could be partially changed by raising the sucrose concentration in the rooting medium as proposed by Eliasson [58]; the rooting percentage of these shoots increased from 6 to 56% when 1% sucrose was added to the rooting medium. This strongly supports the hypothesis that carbohydrate availability may be a critical factor in root formation.

The two culture systems used in this study differ in the time that plant tissues are in contact with the liquid medium, which is higher in the case of CIS. Excessive water accumulation in bioreactors may result in oxygen depletion in cells, which may lead to hypoxia, whereas the different sucrose treatments can cause osmotic stress and produce reactive oxygen species [24]. Plants have enzymatic and non-enzymatic antioxidant mechanisms for scavenging reactive oxygen species and to protect plant tissues [60]. Phenolics are a major class of non-enzymatic antioxidants and their concentration in explants subjected to different culture conditions can/may provide useful information about its physiological state [61]. In the present study, the chestnut shoots cultured by CIS showed statistically significant higher levels of total phenolic compounds relative to those cultured by TIS: however, the differences were small, and no variation was detected between sucrose treatments. Higher phenolic content has been correlated with a plant’s defensive response to suboptimal conditions either in vivo [61,62,63] or in vitro [64], although not all plants follow this pattern, and high accumulation of phenolic compounds has been reported with the optimal performance of tara microshoots [65]. Similarly, in rose, more phenolics were detected in shoots cultured by TIS, which showed a better performance, than by CIS [66].

The total antioxidant activity of chestnut shoots was significantly affected by the culture system and the sucrose supplementation. In both systems, the highest antioxidant activity was detected in the treatments with the lowest sucrose concentration, but shoots cultured in CIS showed more antioxidants than those cultured in TIS. High antioxidant activity has been linked with plants subjected to stresses such as hyperhydricity [67,68,69] and high saline concentration [70,71]. The antioxidant activity of chestnut shoots cultured by CIS seems to be negatively correlated with the sucrose supplementation and with morphological parameters of these shoots; the highest antioxidant content corresponded to the smaller shoots (those cultured with low sucrose), suggesting they suffer higher stress levels than those grown with more sucrose. However, in TIS the antioxidant levels did negatively correlate with sucrose concentration but not with growth responses, as all sucrose treatments produced shoots that were similar in number and length but differing significantly in the antioxidant activity. Although we cannot disregard the possibility that in TIS the high antioxidant activity in treatments with reduced sucrose could be related to a physiological stress that was not reflected in the growth response, TIS seems to be the method of choice to culture chestnut shoots with reduced sucrose concentrations. In the present study, the shoots cultured by TIS with 0.5% sucrose showed suitable proliferation and rooting responses, as well as an accumulation of monosaccharides similar to the other treatments. These results suggest TIS-grown shoots may have developed a higher photosynthetic activity than the shoots cultured by CIS with the same sucrose supplementation. This could enhance their capacity for photoautotrophic micropropagation.

All other experimental factors being considered equal, the reduced growth provided by CIS compared to TIS appears to be related to some specific characteristics of the culture environment. It is postulated that a higher stress is caused by the continuous immersion of the basal sections in liquid medium, but other factors including vessel size, could be important as a bigger volume could prevent an adequate gaseous exchange and lead to suboptimal CO_2_ (adversely affecting photosynthesis) or supra-optimal ethylene concentration (influencing stress). Future investigations aimed at ascertaining the critical factors limiting the use of CIS containers will include the study of the dynamics of gases inside vessels of different sizes and with a different number of gas inlets–outlets, as well as the photosynthetic activity of plant tissues during shoot development.

## 4. Materials and Methods

### 4.1. Plant Material and Standard Culture Conditions

The chestnut genotypes used in this study, named CO53 and PO42, were selected for their tolerance to ink disease during a survey carried out in natural stands of Galicia (NW Spain) by the Agroforestry Department of the company Transformación Agraria SA (TRAGSA) between 2000 and 2005. CO53 and PO42 were characterized as natural hybrids of *C. sativa* and *C. crenata* [15] and were established in vitro from basal sprouts as described in Vidal et al. [72]. Stock cultures were maintained in test tubes containing GD (Gresshoff and Doy 1972), [73] medium supplemented with 0.1 mg/L N^6^ benzyladenine (BA), 3% sucrose and 0.7% (*w*/*v*) Bacto agar. Media were autoclaved at 120 °C for 20 min after adjustment of the pH to 5.7. The cultures were maintained under a 16-h photoperiod provided by cool-white fluorescent lamps (photosynthetic photon flux density (PPF) from 50–60 µmol m^−2^s^−1^) at 25 °C light/20 °C dark (standard conditions). Shoots were subcultured every 5 weeks.

### 4.2. Culture in Liquid Medium

Apical sections of chestnut (25–30 mm, with 4 expanded leaves) were cultured in liquid MS (Murashige and Skoog) [74] salt and vitamin mixture with half strength nitrates (MS-½N) and 0.05 mg/L BA supplemented with 0.5%, 1% or 3% sucrose. Clone PO42 was cultured by TIS in commercial Plantform™ bioreactors (www.plantform.se, accesed on 31 March 2022; Plantform, Hjärup, Sweden) as described in [27], whereas the clone CO53 was cultured by CIS in in-house 6-L bioreactors prepared as described by Cuenca et al. [28] and designated as C6. Twenty-four explants were cultured per Plantform™ and 32 per C6. In both cases, 1 cm^3^ rockwool cubes (Grodan, Roermond, The Netherlands) were used as support material. For continuous immersion, the cultures received forced aeration for 1 min at a frequency of 16 times every 24 h. For temporary immersion, the cultures were immersed for 1 min every 4 h and received additional aeration of the same duration and frequency than for continuous immersion. The explants were cultured in an experimental unit designed for our first experiments in chestnut micropropagation with reduced sucrose [33]. In this experimental unit, designated as PAM, the cultures grew under high PPF (150 µmol m^−2^ s^−1^), and CO_2_-enriched air (≈2000 ppm) was injected to the bioreactors during aeration. The photoperiod and temperature regime were the same as under standard conditions. After 8 weeks of culture, morphological data were recorded and shoots were used for rooting or biochemical analysis.

### 4.3. Rooting

For root induction, vigorous shoots of 30–40 mm height (Figure 9a) were inoculated in 300 mL jars with 50 mL of half strength MS (MS^1/2^) supplemented with 25 mg L^−1^ indole-3-butyric acid (IBA), 0.7% Bacto agar and the same sucrose concentration that was present in the proliferation medium (Figure 9b,d). After 24 h, shoots were transferred to IBA-free medium for rooting expression and distributed between three sucrose treatments. Shoots were inserted in rockwool cubes (2 cm of side) soaked in liquid MS^1/2^ with 0.5, 1 or 3% sucrose (Figure 9c) and introduced in Plantform™ bioreactors without the inner baskets (Figure 9e,f). Plantforms^™^ were aerated for 1 min at a frequency of 3 times every 24 h for 5 weeks.

### 4.4. Biochemical Quantifications

#### 4.4.1. Monosaccharides Content

The method used for the quantification of monosaccharides in chestnut explants was the dinitrosalicylic acid (DNS) method [75,76]. Briefly, 100–150 mg of apical leaves were homogenized with 2 mL of distilled water and centrifuged for 5 min at 10,000 g. The supernatant was collected, mixed with DNS, heated in a thermoblock at 100 °C for 5 min and placed on ice for 5 min before quantitation in a spectrophotometer at 540 nm.

#### 4.4.2. Total Soluble Phenolic Compounds

The extraction was performed according to Díaz et al. [62]. Individual shoots were collected and leaves were homogenized with methanol 80%. The mixture was centrifuged at 10,000 g for 5 min and the supernatant was used for the analysis of soluble phenolic compounds and antioxidants.

For total soluble phenolic compounds, the Folin–Ciocalteu method was used [77]. The content of the soluble phenols was calculated from a standard curve obtained with different concentrations of gallic acid. The results were expressed as gallic acid mg equivalents/g on a fresh weight basis.

#### 4.4.3. Antioxidant Activity

The extraction was performed as described for soluble phenols. Antioxidant activity of plant extracts was determined through spectrophotometry using a 1,1-diphenyl-2-picrylhydrazyl (DPPH) scavenging radical assay [78], and the results were expressed as TROLOX mM equivalents per g on a fresh weight basis.

### 4.5. Data Recording and Statistical Analysis

The parameters analyzed were: (a) the number of normal shoots longer than 15 mm produced by each explant (NS); (b) the length of the longest shoot per explant (SL); (c) the length (LL) and width (LW) of the largest leaf per explant; (d) rooting percentage; (e) monosaccharides; (f) total soluble phenolic compounds; (g) total antioxidant activity.

For morphological parameters, data comprised two replicates per treatment and 32 explants per replicate for shoots cultured in CIS, and three replicates with 24 explants each for shoots cultured in TIS. For rooting experiments and for biochemical analysis, data corresponded to 32 and 24 shoots per treatment, respectively. Data were analysed by Levene’s test (to verify the homogeneity of variance), and subsequently were subjected to analysis of variance (ANOVA) followed by the comparison of group means (Tukey-b test), or to the Welch ANOVA followed by Games–Howell post hoc comparison when heteroscedasticity was detected. For two-way ANOVA, when an interaction between two factors was observed, Bonferroni’s adjustment was applied to detect the simple main effects in multiple post hoc comparisons. Data of rooting percentages were analyzed by Pearson’s Chi-square test. Statistical analyses were performed using SPSS 26.0 (IBM).

## Figures and Tables

**Figure 1 plants-11-00965-f001:**
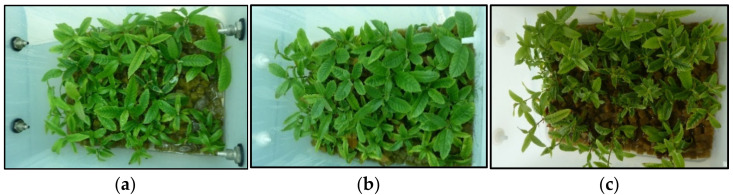
Chestnut shoots of clone CO53 after 8 weeks of culture by continuous immersion with 0.5% sucrose (**a**), 1% sucrose (**b**) and 3% sucrose (**c**).

**Figure 2 plants-11-00965-f002:**
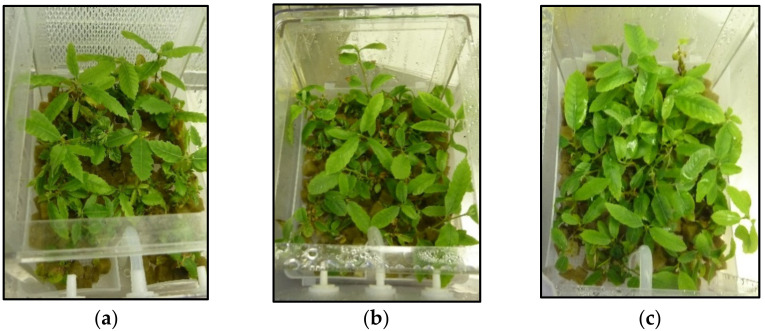
Chestnut shoots of clone PO42 after 8 weeks of culture by temporary immersion with 0.5% sucrose (**a**), 1% sucrose (**b**) and 3% sucrose (**c**).

**Figure 3 plants-11-00965-f003:**
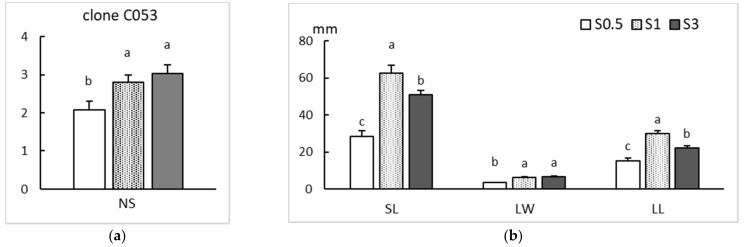
Effect of sucrose supplementation (0.5, 1 and 3%) on proliferation rates of apical sections of clone CO53 shoots cultured by continuous immersion. (**a**) Number of shoots (NS). (**b**) Length of the longest shoot (SL) and length (LL) and width (LW) of the largest leaf per explant. Values are the mean ± standard error from two replicate trials each with 2 bioreactors (64 explants/treatment). For each variable, different letters indicate significant differences at *p* < 0.05.

**Figure 4 plants-11-00965-f004:**
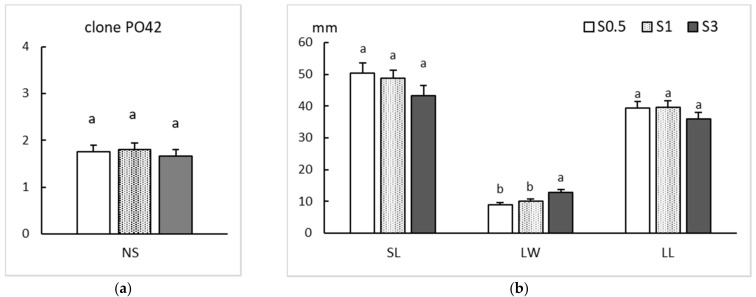
Effect of sucrose supplementation (0.5, 1 and 3%) on proliferation rates of apical sections of clone PO42 shoots cultured by temporary immersion. (**a**) Number of shoots (NS). (**b**) Length of the longest shoot (SL) and length (LL) and width (LW) of the largest leaf per explant. Values are the mean ± standard error from three replicate trials each with 2 bioreactors (72 explants/treatment). For each variable, different letters indicate significant differences at *p* < 0.05.

**Figure 5 plants-11-00965-f005:**
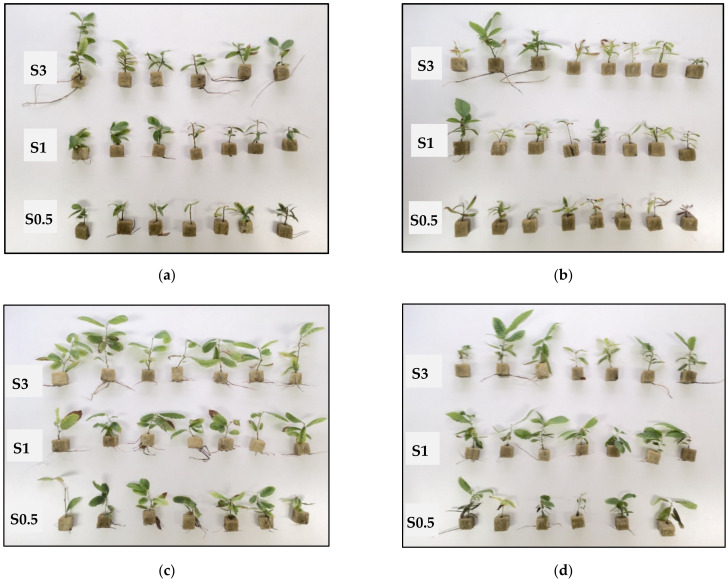
Chestnut shoots from clones CO53 and PO42 multiplied and rooted with different sucrose concentrations. (**a**) Shoots of CO53 cultured in sucrose 3% and rooted in sucrose 0.5, 1 and 3%. (**b**) Shoots of CO53 cultured in sucrose 0.5% and rooted in sucrose 0.5, 1 and 3%. (**c**) Shoots of PO42 cultured in sucrose 3% and rooted in sucrose 0.5, 1 and 3%. (**d**) Shoots of PO42 cultured in sucrose 0.5% and rooted in sucrose 0.5, 1 and 3%. Photos taken 5 weeks after root induction. S0.5, S1 and S3: Sucrose 0.5, 1 and 3%.

**Figure 6 plants-11-00965-f006:**
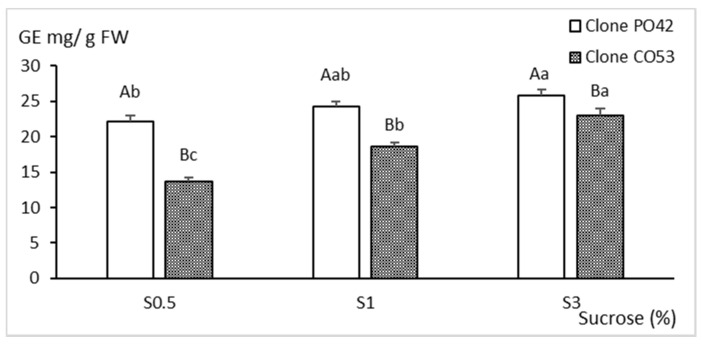
Effect of sucrose supplementation on monosaccharide content of shoots of clones PO42 and CO53 cultured by temporary and continuous immersion, respectively. Values are the mean ± standard error from 24 shoots analyzed independently. Different capital letters indicate significant differences in relation to the genotype, and different lowercase letters indicate significant differences in relation to the sucrose supplementation (*p* < 0.05). GE, Glucose equivalents.

**Figure 7 plants-11-00965-f007:**
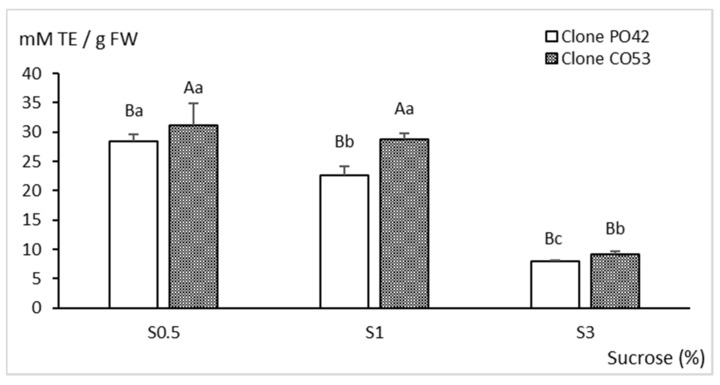
Effect of sucrose supplementation on antioxidant activity of shoots of clones PO42 and CO53 cultured by temporary and continuous immersion, respectively. Values are the mean ± standard error from 24 shoots analyzed independently. Different capital letters indicate significant differences in relation to the genotype, and different lowercase letters indicate significant differences in relation to the sucrose supplementation (*p* < 0.05). TE: Trolox equivalents.

**Figure 8 plants-11-00965-f008:**
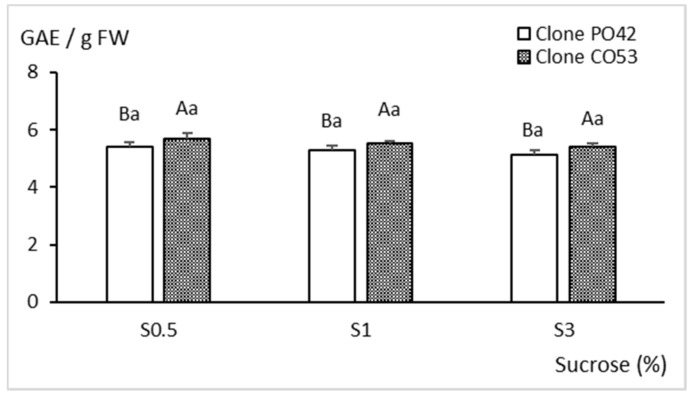
Effect of sucrose supplementation on soluble phenolic compounds of shoots of clones PO42 and CO53 cultured by temporary and continuous immersion, respectively. Values are the mean ± standard error from 24 shoots analyzed independently. Different capital letters indicate significant differences in relation to the genotype, and different lowercase letters indicate significant differences in relation to the sucrose supplementation (*p* < 0.05). GAE: Gallic acid equivalents.

**Figure 9 plants-11-00965-f009:**
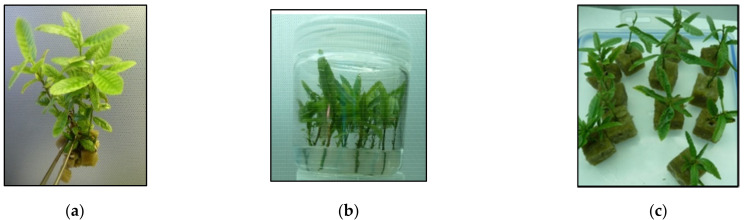
Rooting of chestnut shoots from clones CO53 and PO42. (**a**) Shoots of CO53 cultured in S3% selected for rooting. (**b**,**c**) Shoots of CO53 cultured in S1% during root induction (**b**) and inserted in rockwool cubes. (**c**,**d**) Shoots of PO42 cultured in S1% during root induction. (**e**,**f**) Shoots of CO53 cultured in S1% and transferred to medium with S0.5, 1 and 3% 24 h after root induction.

**Table 1 plants-11-00965-t001:** Effect of sucrose supplementation on the rooting percentage of chestnut. Shoots of clones PO42 and CO53 were cultured by temporary and continuous immersion with 0.5, 1 and 3% sucrose. Shoots from each sucrose treatment during the multiplication phase were distributed between 0.5, 1 and 3% sucrose treatments during the rooting expression phase. Means ± standard error were calculated from 2 replicates, each with 16 shoots per treatment. For each clone, different uppercase letters within a column indicate significant differences regarding sucrose concentration during multiplication, and different lowercase letters within a row indicate significant differences regarding sucrose concentration during rooting expression (Pearson’s Chi-square test, *p* < 0.05).

Clone	Sucrose During Multiplication (%)	Sucrose on the Rooting Medium (%)		
		0.5	1	3
PO42	0.5	66 ± 22.1 Aa	72 ± 22.1 Ba	56 ± 8.8 Ba
	1	75 ± 17.7 Aa	78 ± 13.3 Ba	75 ± 26.5 Ba
	3	91 ± 13.3 Aa	97 ± 4.4 Aa	100 ± 0.0 Aa
CO53	0.5	6 ± 8.8 Ab	56 ± 26.5 Aa	44 ± 17.7 Ba
	1	13 ± 17.7 Ab	59 ± 13.3 Aa	56 ± 17.7 Ba
	3	25 ± 8.8 Ac	59 ± 13.3 Ab	84 ± 4.4 Aa

## Data Availability

Not applicable.

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
