# Peer review of "Effect of Sucrose on Growth and Stress Status of Castanea sativa x C. crenata Shoots Cultured in Liquid Medium"

_plants, 2022, doi:10.3390/plants11070965_

Round 1

Reviewer 1 Report

Figure 1 and 2 showed color of shoots. Why the authors do not  measure the chlorophyll content? leaves in  the figure 2 c are yellow in the margins . how do you explain this?

 Figures  3 and 4 : Length of the longest shoot (SL) and length (LL) and width (LW) of the largest leaf per explant? but the average length of the other shoots  of same explants?

figure 5 :  the figure is expressed as Rooting expression medium , without any significant statistical difference. please transform in table and show the SE. 32 shoots per treatment without difference is very strange.

Author Response

We appreciate the comments of the Reviewer 1 for improving the manuscript.

Figure 1 and 2 showed color of shoots. Why the authors do not  measure the chlorophyll content?

In Figures 1 and 2 we wanted to show the general appearance of the shoots, including the color. With respect to the chlorophylls, we did some measurements in this plant material during our first investigation in the effect of sugar in chestnut propagation. That report is cited in the manuscript: Vidal et al, 2017 (Proliferation and rooting of chestnut under photoautotrophic conditions. In Proceedings of the 4th International Conference of the IUFRO Unit 2.09.02 on “Development and application of vegetative propagation technologies in plantation forestry to cope with a changing climate and environment” pp. 119–127). We reported no correlation between chlorophylls and growth responses. Similar results had been reported for other plants such as Vernonia condensata (Fortini et al. 2021; doi:10.1007/s11240-020-01981-5), myrtle (Lucchesini et al 2006, doi:10.1007/s10535-006-0001-9], apple [Kim et al, 2020; doi:10.1007/s13580-020-00257-3), tobacco, potato, strawberry and rapeseed (Ševčíková et al 2019 doi:10.1111/ppl.12893) and also chestnut (Sáez et al. 2012; doi:10.1016/j.scienta.2012.02.005). For all these reasons in this study we decided to focus on other biochemical parameters instead on photosynthetic pigments, which was not adding to our current understanding.

leaves in  the figure 2 c are yellow in the margins . how do you explain this?

The clearer zones in some leaves of this figure are due to light reflection when we took the picture in the flow cabinet.

 Figures  3 and 4 : Length of the longest shoot (SL) and length (LL) and width (LW) of the largest leaf per explant? but the average length of the other shoots  of same explants?

Data on length and width of leaves refers to the largest leaf of each explant, as explained in the figures and in the materials and method section (Data recording and statistical analysis).

figure 5 :  the figure is expressed as Rooting expression medium , without any significant statistical difference. please transform in table and show the SE. 32 shoots per treatment without difference is very strange.

These data appear now in Table 1 and letters for significant differences have been added as suggested.

Reviewer 2 Report

Comments to authors:

Thank you for inviting me the paper titled: Effect of Sucrose on Growth and Stress Status of Castanea sativa x C. crenata Shoots Cultured in Liquid Medium

The authors have good experience in their work. I've detected some flaws and gaps. In my opinion the main problem of research is regarding the use of only one genotype for one of the 2 different culture systems. Under these settings, some of the statements and conclusions could be biased by the use of different conditions for 2 genotypes. For example, from line 154 to 156 was affirmed that "In the clone CO53 less shoots formed roots, percentages ranging from 6 to 84%. The lowest response (6%) corresponded to shoots proliferated with 0.5% sucrose and rooted with the same sucrose concentration (Figure 5, Figure 6B). This value contrasts with the response of clone PO42 to the same treatment (66%)." ... and so on. I consider that must be mandatory to explain why TIS was used for a clone, and CIS for the other: the right way had would be assessing both clones cultured with both systems. And the most important, it's not right to compare results that come from different conditions, at least not in the way that it has been done.

I also recommend to do a better literature review about micropropagation of other nut trees and enrich the introduction. For example the authors can address to the following papers:

Asayesh ZM, Vahdati K, Aliniaeifard S (2017) Investigation of physiological components involved in low water conservation capacity of in vitro walnut plants. Scientia Horticulturae. 224: 1-7.

Maleki Asayesh, Vahdati K, Aliniaeifard S, Askari N (2017) Enhancement of ex vitro acclimation of walnut plantlets through modification of stomatal characteristics in vitro. Scientia Horticulturae. 220: 114-121.

Author Response

We appreciate the comments and suggestions of the Reviewer 2 for improving the manuscript.

Thank you for inviting me the paper titled: Effect of Sucrose on Growth and Stress Status of Castanea sativa x C. crenata Shoots Cultured in Liquid Medium

The authors have good experience in their work. I've detected some flaws and gaps. In my opinion the main problem of research is regarding the use of only one genotype for one of the 2 different culture systems. Under these settings, some of the statements and conclusions could be biased by the use of different conditions for 2 genotypes. For example, from line 154 to 156 was affirmed that "In the clone CO53 less shoots formed roots, percentages ranging from 6 to 84%. The lowest response (6%) corresponded to shoots proliferated with 0.5% sucrose and rooted with the same sucrose concentration (Figure 5, Figure 6B). This value contrasts with the response of clone PO42 to the same treatment (66%)." ... and so on.

The authors acknowledge the reviewer comment. The text of lines 154 and thereafter has been modify to clarify that the culture system was different in both genotypes.

I consider that must be mandatory to explain why TIS was used for a clone, and CIS for the other: the right way had would be assessing both clones cultured with both systems. And the most important, it's not right to compare results that come from different conditions, at least not in the way that it has been done.

We understand the reviewer’s concern. We could have performed TIS and CIS with a single clone, but we wanted to extend the scope of the study. In previous reports (cited in this paper with numbers 27, 28 and 33) we had worked with the genotypes we used in the study, among others, as we explained in lines 232-238 of the discussion section. For more than 10 years we multiplied these clones in liquid medium and found that their responses to TIS and CIS were very similar. Besides, we carried out some experiments with different sucrose concentrations with both clones and the two systems, as suggested by the reviewer, and found them to be indistinguishable between the 2 genotypes. Some of these experiments are described in the reference number 34 (Vidal et al, 2017) but the number of bioreactors that we would need at the same time for setting a proper experiment comparing the two systems with the two clones and the three sucrose concentrations and the corresponding replicates exceeded the capacity of the PAM experimental unit we used in the study.  

I also recommend to do a better literature review about micropropagation of other nut trees and enrich the introduction. For example the authors can address to the following papers:

Asayesh ZM, Vahdati K, Aliniaeifard S (2017) Investigation of physiological components involved in low water conservation capacity of in vitro walnut plants. Scientia Horticulturae. 224: 1-7.

Maleki Asayesh, Vahdati K, Aliniaeifard S, Askari N (2017) Enhancement of ex vitro acclimation of walnut plantlets through modification of stomatal characteristics in vitro. Scientia Horticulturae. 220: 114-121.

We appreciate the suggestion of the Reviewer 2. We have read the papers but, while being interesting, are actually different to this specific manuscript on chestnut culture on bioreactors, as they aren't focus on bioreactors, on sucrose effect, on liquid medium or on stress, or measure any of the parameters we study. For this reason, we will consider including them in our next paper focused on changes on chestnut plantlets during acclimation.

Reviewer 3 Report

The authors studied the potential benefits of decreasing sucrose supplementation on growth, rooting, and physiological status (monosaccharide content, soluble phenolics, and antioxidant activity) of chestnut shoots of clone CO53 and PO42 cultured in liquid medium by continuous and temporary immersion system, respectively. The chestnut genotypes CO53 and PO42 were characterized as natural hybrids of Castanea sativa and C. crenata and were selected for their tolerance to ink disease. To compensate the effects of reduced sucrose supply on plant metabolism, explants were cultured with high light intensity and CO2-enriched air in bioreactors with forced ventilation. On the basis of the obtained results, the authors concluded that a temporary immersion system seems to be the method of choice to culture chestnut shoots with reduced sucrose concentrations. This study contributes to the elucidation of the most suitable conditions for the micropropagation of disease-resistant chestnut genotypes.

My main question is what was the purpose of using two different clones, CO53 and PO42, cultured by continuous and temporary immersion, respectively? If one would like to study the differences in the response of sugar supplementation related to the culture system (TIS or CIS) isn’t it better to use the same genotype.

Minor remarks:

Line 114 – “Typical shoot growth from clones CO53 and PO42 cultured in liquid medium supplemented with 0.5, 1 and 3% sucrose is shown in Figure 2; Figure 3 (below)” – These data are presented in Figure 1 and Figure 2.

Line 220 – “In a recent study we succesfully micropropagated willow under high PPF and without sucrose when we used bioreactors ventilated with CO2-enriched air instead of jars without ventilation [40], and to obtain good proliferation in chestnut we applied the same principle, adding CO2-enriched air every 90 min to the bioreactors with”

Line 277 – “Higher phenolic content of has been correlated with plant defensive response to suboptimal conditions either in vivo [61-63] or in vitro [64] …”

Author Response

The authors studied the potential benefits of decreasing sucrose supplementation on growth, rooting, and physiological status (monosaccharide content, soluble phenolics, and antioxidant activity) of chestnut shoots of clone CO53 and PO42 cultured in liquid medium by continuous and temporary immersion system, respectively. The chestnut genotypes CO53 and PO42 were characterized as natural hybrids of Castanea sativa and C. crenata and were selected for their tolerance to ink disease. To compensate the effects of reduced sucrose supply on plant metabolism, explants were cultured with high light intensity and CO2-enriched air in bioreactors with forced ventilation. On the basis of the obtained results, the authors concluded that a temporary immersion system seems to be the method of choice to culture chestnut shoots with reduced sucrose concentrations. This study contributes to the elucidation of the most suitable conditions for the micropropagation of disease-resistant chestnut genotypes.

My main question is what was the purpose of using two different clones, CO53 and PO42, cultured by continuous and temporary immersion, respectively? If one would like to study the differences in the response of sugar supplementation related to the culture system (TIS or CIS) isn’t it better to use the same genotype.

We appreciate the reviewer’s comments about our study. Regarding the use of two clones in different systems, at first we had thought that we could carry out TIS and CIS with a single clone, as the number of bioreactors that we would need at the same time for setting a proper experiment comparing the two systems with the two clones and the three sucrose concentrations and the corresponding replicates exceeds the capacity of the PAM experimental unit we used in the study. However, we wanted to extend the scope of the study. In previous reports (cited in this paper with numbers 27, 28 and 33) we had worked with the genotypes we used in the study, among other clones, as we explained in lines 232-238 of the discussion section. For more than 10 years we multiplied these clones in liquid medium and found that their responses to TIS and CIS were very much the same. Besides, we carried out some experiments with different sucrose concentrations with both clones and the two systems, and found a similar behavior among the 2 genotypes. Some of these experiments can be found reported in the reference number 34 (Vidal et al, 2017).

Minor remarks:

Line 114 – “Typical shoot growth from clones CO53 and PO42 cultured in liquid medium supplemented with 0.5, 1 and 3% sucrose is shown in Figure 2; Figure 3 (below)” – These data are presented in Figure 1 and Figure 2.

Thank you, it was a mistake and has been mended.

Line 220 – “In a recent study we succesfully micropropagated willow under high PPF and without sucrose when we used bioreactors ventilated with CO2-enriched air instead of jars without ventilation [40], and to obtain good proliferation in chestnut we applied the same principle, adding CO2-enriched air every 90 min to the bioreactors with”

Thank you, it was a mistake, the last “with” has been deleted.

Line 277 – “Higher phenolic content of has been correlated with plant defensive response to suboptimal conditions either in vivo [61-63] or in vitro [64] …”

 Thank you, it was a mistake, the last “of” has been deleted.